Employing deep learning in crisis management and decision making through prediction using time series data in Mosul Dam Northern Iraq

Khafaji Khalid MK khalid.alkhafaji@enetcom.u-sfax.tn 1
Ben Hamed Bassem 1
National School of Electronics and Telecommunications, University of Sfax , Sfax , Tunisia
Balas Valentina Emilia
Electronic publication date: 2024 Oct 31
Publication date: 2024
Volume: 10
Electronic Location ID: e2416
Received 2024 Apr 29; Accepted 2024 Sep 24
Copyright: ©2024 Khafaji and Ben Hamed
Copyright year: 2024
Copyright holder: Khafaji and Ben Hamed
License: This is an open access article distributed under the terms of the Creative Commons Attribution License, which permits unrestricted use, distribution, reproduction and adaptation in any medium and for any purpose provided that it is properly attributed. For attribution, the original author(s), title, publication source (PeerJ Computer Science) and either DOI or URL of the article must be cited.
License URL: https://creativecommons.org/licenses/by/4.0/

Keywords: Prediction, Time series, Deep learning, Flood, Crisis, Decision

Funding: The authors received no funding for this work.

==============================
Specifically, Iraq is threatened in its second-largest northern city, Mosul, by the collapse of the Mosul Dam due to problems at the root of the dam, causing a wave of floods that will cause massive loss of life, resources, and public property. The objective of this study is to effectively monitor the level of dam water by predicting the level of water held by the dam In anticipation of achieving flood stage and breaking the dam, and supporting its behavior through formation 14-day time series data to predict seven days later. Used six deep learning models (deep neural network (DNN), convolutional neural network (CNN), convolutional neural network long short-term memory (CNN-LSTM), CNN-LSTM-Skip and CNN-LSTM Skip Attention) that models were trained to predict the water level in the dam; these levels of being under surveillance and prepared In case of danger, alert people to potential flood threats depending on the dam’s water level. These time series were created from the actual data sets of the dam; it’s a fundamental historical reading for 13 years (1993–2006) of the water level stored in the Mosul dam and was adopted in coordination with the Iraqi Ministry of Water Resources and the Centre for Research on Dams and Water Resources at Mosul University. The methodology applied in this study shows the model’s performance efficiency and the prediction results’ low error rate. It also compares those practical results to determine and adopt the performance-efficient model and a lower error rate. The comparison of these results proved the accuracy of its results, and superior to the CNN-LSTM model, it has the highest ability to perform through high accuracy with MAE = 0.087153 and time steps = 0 s 196 ms/step and loss = 0.00067. The current study demonstrated the ability to predict the water level in Mosul Dam, which suffers from foundation problems and may collapse in the future. Therefore, the water level in the dam must be monitored accurately. It also aims to test the effectiveness of the six models proposed in this study after evaluating their performance and applying the prediction process within a scenario to obtain predictive values after 14 days. The results showed the practical effectiveness of the hybrid CNN-LSTM model in correctly and accurately obtaining predictive values within the integrated framework of the required scenario. The study concluded that it is possible to enhance the ability to monitor and identify the potential risk of Mosul Dam at an early stage, and it also allows for proactive crisis management and sound decision-making, thus mitigating the adverse effects of crises on public safety and infrastructure and reducing human losses in areas along the Tigris River.

Introduction

Dam management relies heavily on accurate readings of water storage and release, as well as the causes that threaten the dam body, the repercussions of which are hazardous to lives, property, lands, institutions, the economy...etc., which may cause a major crisis that is difficult to confront or recover from Angelakis et al. (2024).

Specifically, Iraq is threatened in its second-largest northern city, Mosul, by the collapse of the Mosul Dam due to problems with the dam’s foundations. This would cause a wave of floods that would cause massive losses in lives, resources, and public property (Al-Taiee & Mustafa, 2021; Munawar et al., 2021).

The goal of this study is to effectively monitor the water level of Mosul Dam by predicting the level of water held by the dam in anticipation of reaching the flood stage and breaking the dam and to support its behavior by generating 14-day time series data to determine the water level in the dam. The future values were predicted for the next seven days. We used six deep learning models (deep neural network (DNN), convolutional neural network (CNN), convolutional neural network long short-term memory (CNN-LSTM), CNN-LSTM-Skip and CNN-LSTM Skip Attention) that trained the models to predict the water level in the dam. These levels are monitored and prepared in case of danger, alerting people to potential flood threats depending on the dam water level. These time series were generated from actual dam datasets. It is an essential historical reading for 14 years (1993-2006) of the water levels stored in the Mosul Dam. It was approved in coordination with the Iraqi Ministry of Water Resources and the Dams and Water Resources Research Center at the University of Mosul. The results of the methodology applied in this study show the efficient performance of the model and the low error rate in the prediction results, and the practical results are compared to determine and adopt a model with efficient performance and a lower error rate. Comparison of these results proved the accuracy of its results and its superiority to the CNN-LSTM model, as it has the highest performance ability through high accuracy with MAE = 0.087153, time steps = 0s 196 ms/step, and loss = 0.00067.

A data scientist’s value lies in their ability to deal with many variables simultaneously (Munawar et al., 2022). In significant data analysis, time series prediction is crucial. Good generalization from limited data is essential for predicting noisy, nonstationary, and disordered time series. Time series prediction models benefit significantly from sparsity since it shortens the duration of the review process. This becomes more crucial when the models are utilized as part of trade flows, where instantaneous evaluation is essential (Munawar et al., 2021).

The Mosul Dam was taken as a case in this study due to the importance of this dam in Iraq and the dangerous condition it carries in its foundations, which is considered a critical threat to Iraq (Lei et al., 2021). Many breach parameter prediction techniques used in risk assessment studies of embankment dam collapse are based on research into past dam failures. In a similar vein, peak breach outflow may be predicted with the use of relations derived from case study data (Jaseena & Kovoor, 2022).

This study aims to employ deep learning models and find the most efficient and accurate model to benefit from its results in applying proactive crisis management in the Mosul Dam (and other dams, if possible) and making sound decisions. Therefore, this practical study will contribute to mitigating the adverse effects of the Mosul Dam on public safety and infrastructure. Reducing human damage and expected losses in areas along the Tigris River. This study seeks to emphasize the importance of proactive management of dams, given their importance and danger at the same time, through interdisciplinary cooperation, integration of scientific knowledge, and technological development in the field of artificial intelligence in crises and crises. Risk management processes in dams through data analysis and accurate forecasting using neural networks and related deep learning models. This integration is critical for developing effective crisis management systems and applying early warning measures to mitigate the risk of dam failure and its impact in areas downstream of the dam basin. Therefore, it is proposed that this methodology be applied in the future in disaster management and decision-making to prevent risks and their effects on the Mosul Dam.

Related Works

Chen et al. (2023) discuss an advanced hydrological-geotechnical model for prediction, enriching the discussion on prediction accuracy and model efficiency. Zhang et al. (2023) and Zhao et al. (2024) explore precipitation processes and mechanisms, adding depth to the understanding of regional climatic factors relevant to the Mosul Dam scenario. Du & Wang (2013) focus on spatial correlations for seismic activities, enhancing the geotechnical aspects of the model. Zhu (2023) introduces an adaptive decision model using deep reinforcement learning, potentially improving predictive accuracy. Ye et al. (2024) provide insights into long-term deformation monitoring, aligning with the article’s focus on the Mosul Dam. Liu et al. (2024) research fault diagnosis using time series segmentation, which can inspire similar techniques for hydrological models. Hu et al. (2024) predict surface settlement using machine learning, introducing additional methods to enhance predictive capabilities. Yang et al. (2022) discuss graph convolutional network, applicable to spatial–temporal analysis in the article. Chen et al. (2024) use fractal theory and machine learning for metallogenic prediction, offering novel perspectives for better prediction outcomes. Yin et al. (2023) analyze precipitation and river discharge during the operation of the Three Gorges Dam, providing a basis for comparative analysis. Qin et al. (2024) introduce an RCLSTMNet for forecasting, potentially enhancing the hybrid CNN-LSTM models discussed in the article. By integrating these references, the article will benefit from a richer literature foundation, reinforcing the validity and relevance of its findings within the broader context of current research. Dayal et al. (2024) local variations in weather conditions in space and time and complex physical processes lead to great difficulty in predicting reservoir water levels. The typical man’s need for water management today demands, in any case, the provision of a reliable and accurate forecast of dam water levels. This article captures the potential use of LSTM networks on daily weather data from reservoirs. Their performance is compared among these seven high-performance models (M1–M7), trained on different window sizes and horizons with those data. According to the performance metrics, Model M7 performs best in predicting reservoir water levels, with an R2 value of 0.93, RMSE of 2.94, and MAPE of more than 0.01. Yadav & Mathur (2020) dam management and flood management rely heavily on precise prediction of water storage and release. The model applied to the prediction process determines how well the water level may be predicted. Water management issues, including drought performance, flood control, water supply, and irrigation, all rely on accurate predictions of future discharge levels or other measurable water quantities. Won et al. (2022) describe the system used to predict and issue warnings about urban flooding in South Korea’s high-risk regions. This method was created using a combination of a deep learning model and a runoff model from rainfall. Model-driven construction was used to create the accurate physical model, which included integrated inland river and flood control systems such as pump stations and underground storage. Information collected in August 2020 from measuring and pumping stations along a stream in a city was used to calibrate the rainfall-runoff model, and the resulting R2 value was given. The model-driven approach studied how the urban river’s flood detection rules would change with different rainfall scenarios. Long short term memory (LSTM) and basic artificial neural network (ANN) are both examples of deep learning models. Stack LSTM and bidirectional LSTM were created to both predict and caution of urban river floods. The potential for floods in an urban river was employed to instruct deep learning models using 10-minute hydrological datasets from water stations. When applied to water level predictions with a lead time of 30 min, a prediction and warning approach based on the bidirectional LSTM had an R2 value of 0.9, which suggests the potential for effective flood predictions and alerts. The researchers of this case study intend to utilize their findings to enhance flood prevention and safety in the most vulnerable regions of South Korea. It is intended that the indications provided by the deep learning models in this study will be utilized, however, additional efforts must be made to expand the deep learning model to other metropolitan areas. Hou et al. (2021) give an experimental dataset for the spread of flooding along a river channel; these measurements may be used to check the accuracy of computer simulations of floods on rivers. Records of the spatial evolution of flooded regions and water levels at gauges of relevance during the propagation process make up the available measures of flood propagation. The model’s flexibility and ease of use make it a promising candidate for predicting and characterizing floods in the future. This method offers a novel and efficient technological strategy for data collection in experimental settings. Cameras struggle to record the changes in the inundated region as water recedes because cement’s hydraulic permeability constrains this method. A combination of side view recordings and inverse perspective transformations in the study area is required to collect representative samples to determine the extent of flooding. Moishin et al. (2021) presented an approach that might be improved to predict flooding events hourly. Such accurate prediction at shorter timescales is bound to provide more time for informed decision-making by governments, organizations, and individuals to be prepared for imminent flood situations; hence, lives and infrastructural resources can be saved. The researchers built a flood prediction model that was developed and evaluated using a hybrid deep learning (ConvLSTM) method that combines the predictive benefits of CNN and LSTM network. Nine separate rainfall datasets from flood-prone areas of Fiji, which suffer from flood-driven destruction practically every year, are used to test the effectiveness of the proposed ConvLSTM model. The results show that the ConvLSTM-based flood model outperforms the benchmark approaches when comparing predictions made for the future 1, 3, 7, and 14 days. In this period of extreme weather occurrences, the findings showed the practical efficacy of ConvLSTM in properly predicting IF and its potential use in disaster management and risk prevention. This study aims to predict the level of water held by the Mosul dam, which has problems with its foundations and may have caused its future collapse, and to test the effectiveness of the six models proposed. The results show that the CNN-LSTM model is superior to the performance of reference approaches when comparing forecasts of 14 days in the future. The results showed the practical effectiveness of CNN-LSTM in accurately and adequately forecasting the integrated framework and its potential use in disaster management and risk prevention. Luo & Mirabbasi (2022) computer hardware is another aspect of the computing environment for deep learning and neural networks. Computing devices such as graphics processing units (GPUs) and tensor processing units (TPUs) play a vital role in supporting neural networks. For example, the NVIDIA V100 GPU is one of the most popular computing devices for supporting neural networks due to its ability to process large amounts of data.

Study Area

Figure 1, represents Iraq’s most giant hydraulic construction, the Mosul Dam. Even though the dam has been in operation since 1986, it was chosen as a case study to foresee flood disasters caused by a potential Mosul Dam failure due to a foundation problem located on the Tigris River in northern Iraq. It is an earth-fill zonal dam with a height of 105 m and a volume capacity of 11.1 billion cubic meters. The current research’s goals were to predict future water levels in the dam and monitor those levels so that, if a critical or dangerous level is expected in the future, the crisis team will be fully informed and work to develop the plans and emergency procedures necessary for combating this crisis that jeopardizes the resources and the population living along the Tigris River’s course. Pertinent public sector organizations may use this data to create strategic plans to lessen the impact on lives lost and assets of the local populace and government in the downstream areas. A study of the Mosul Dam by the US Army Corps of Engineers from June 2004 to July 2007 indicated the risk of the dam failing. When ISIS took control of the dam site region in 2014, the media raised this issue (Ahmed et al., 2010).

Figure 1 Mosul Dam location.

Map credit: Al-Taiee & Mustafa, 2021.

Methodology

Dataset

The historical period (time series) taken in this study is (1993–2006), where the official readings of the (level) parameter for the dam area were recorded on a basis (daily, monthly, and yearly). The dataset was adopted in coordination with the Iraqi Ministry of Water Resources.

Descriptive and quantitative statistics were provided for the (water level) data, as shown in Table 1, which includes 4,748 rows (values), which is the number of values that were provided in the data set adopted in this study for a period of approximately 13 years (1993–2006), and two columns. (0,1) represents the variables (date, attribute) that will be adopted as primary inputs. After cleaning the data from impurities and noise, data = zero missing values. Thus, the data became accurate inputs that did not carry errors or noise.

Table 1 Information of dataset.

COLUM	Name	Missing values	Total rows	% Missing	
0	Date	0	4748	0.0	
1	Level	0	4748	0.0	

Data scale

Since the study aims to employ data within artificial intelligence techniques to help make the right decisions in crisis management processes, the prediction methodology is proposed by deep learning models. The idea here is that the prediction mechanism will depend on the following: It will take 14 days before performing the calculations on it to predict seven days in advance, which means that it will take two weeks to predict the outcome within the next seven days, so will have time to manage our crisis and make the decision because have genuine and new outputs that have not passed an extended period or within a stable and stable path. Those outputs are expected to carry critical values that indicate a state of danger. So, this study will use a function (window-dataset) that will take the data and start converting and will label our data that each row will be the minimum scale X for the last two weeks. Limit the values of our data between (−1 and 1) so that the error rate is not too large and does not overload the dataset within the model.

Data distribution

Figure 2, the box plot for the level variable (water level) and divided into two parts (annual and quarterly) represents the box plot for each year from the beginning of 1993 to 2006, which is the end of our data. The years and all density values are centered between 252 and 256, with a few exceptions such as the tails (noise), and this was clear from the histogram of Fig. 3 in next section.

Data density

Figure 3, is a density-wise display of water level values at Mosul Dam, made by water level mass over time, and values between approximately 252 and 256 or slightly more are shown in the plot and for the period (1993–2006). Other values are tails (noise).

Data splitting

Our first function is split data, which takes data and divides it into three things (training, validation, and testing). The ratio will be divided by 80, 10, 10. That will be 3,398 for practice, 850 for checking, and 500 for testing, as shown in the Table 2.

Figure 2 Data distribution.

Figure 3 Data density.

Table 2 Multivariate dataset.

Train	Validation	Test	
3,398	850	500	

Neural network design of deep learning models

Neural networks, which are based on the composition of our brains, are frequently discussed in deep learning. Every node in the network resembles a human neuron, connecting different brain parts.

The LSTM has been enhanced based on the first RNN. The new recurrent neural network (RNN), an LSTM network, has input and output gates (Yu et al., 2019).

The LSTM maintains the cell’s internal state during the cycle; this state is used to establish temporal connections. After that, it traverses the gate that forgets the input information (Li et al., 2024).

Ct: state of the cell.

W: matrix of weight coefficients.

b bias term.

σ: sigmoid activation function.

tanh represents the hyperbolic tangent activation function (Sherstinsky, 2020).

The input, forget, and output gates are the three separate gates that make up the LSTM model. Equations (1), (2) and (3) may be used to calculate the input gate of a LSTM (Li et al., 2024), where (xt) stands for the current input, (ht) for the current output, and (ht1) for the preceding output. The current and previous cell states are represented by the variables (Ct) and (Ct1), respectively. (1) it=σWi.ht−1,xt+bi,

(2) Ct=tanhWc.ht−1,xt+bc,

(3) Ct=ft∗Ct−1+it∗Ct.

Equations (1) and (2) are employed to determine the specific content that is being stored in the cell state. This is accomplished by applying a sigmoid layer to the prior hidden state (ht1-1 ) and a hyperbolic tangent layer (tanh) to the current input (xt), respectively. The terms “wi” and “bi” are used in academic literature to denote the bias of the input gate and the weight matrix, respectively. By combining the outputs of the sigmoid function with a value of 1 and the hyperbolic tangent function with a value of 2, using a specific mathematical procedure labelled as 3, the unique cellular state is produced. (4) ft=δWf.ht−1,xf+bf.

Equation (4), where Wf represents the weight matrix and bf is the offset, stands for the forget gate. To determine the likelihood of forgetting certain information from the preceding cell, sigmoid and dot products are used (Li et al., 2024). (5) δt=δWo.ht−1,xt+bo,

(6) ht=ttanhct.

Wo and (bo) are the LSTM’s output gate weight and bias, respectively, in Eq. (5). In Eq. (6), the final output is multiplied by the tanh of the state of the recently obtained information (Ct) (Yu et al., 2019). This block takes an input tensor of shape (31,10240) in the aforementioned implementation. Using a single 256-unit LSTM layer, followed by layers for dropout and batch normalization, and then an attention layer, prevented over fitting from occurring.

Definition of models

1. Deep neural network (DNN)

•    Can learn complex patterns in time series data.

•    Can handle large datasets and high-dimensional inputs.

•    Can be used for both univariate and multivariate time series forecasting.

2. Convolutional neural network (CNN)

•   Can capture local patterns and trends in time series data.

•   Can handle high-frequency data and noisy signals.

•   Can be used for anomaly detection and forecasting.

3. Long short-term memory (LSTM)

•   Can capture long-term dependencies and temporal relationships in time series data.

•   Can handle irregular time series data and missing values.

•   Can be used for forecasting and anomaly detection.

4. Convolutional neural network long short-term memory (CNN-LSTM)

•   Combines the strengths of CNNs and LSTMs to capture both local and long-term patterns.

•   Can handle high-frequency data and noisy signals.

•   Can be used for forecasting and anomaly detection.

5. CNN-LSTM Skip model

•   Adds a skip connection to the cnn_lstm model, allowing the model to capture both local and long-term patterns more effectively.

•   Can handle high-frequency data and noisy signals.

•   Can be used for forecasting and anomaly detection.

6. CNN-LSTM-Skip attention model

•   Adds an attention mechanism to the cnn_lstm_skip model, allowing the model to focus on specific parts of the input sequence.

•   Can capture complex patterns and relationships in time series data.

•   Can be used for forecasting and anomaly detection.

Generation of prediction

The essential operation of the predictive model is to provide predictions about water levels for the following 14 days. These predictions are crucial for sending timely alerts and caution to vulnerable locations. The main factor of this study is the water levels in the Mosul Dam for 13-year historical values, which the learning models continually analyze. The warning system provides warnings based on a prediction of water levels, including information on the anticipated elevation changes, the anticipated time of impact, and the strength of the flood wave. Authorities and citizens alike must have access to these alerts to fully comprehend the dangers they face and take the necessary precautions to protect themselves and their communities.

Application of models

Model improvement and evaluation over time

Continuous review and improvement are essential for ensuring the validity and efficacy of the prediction model. The model’s performance, including its accuracy and dependability, will be regularly evaluated using historical data and current observations. Through model updates and recalibration, any possible flaws or areas for improvement may be found throughout this review process and fixed. In conclusion, decision-makers now have a valuable tool for anticipating and addressing any future crises brought on by the Mosul Dam, thanks to the predictive model’s inclusion into the crisis management framework. Authorities may create crisis management plans, distribute resources wisely, and guarantee prompt evacuation and communication using the 14-day water level estimates. This will eventually save lives and have a minimal impact on the impacted people and infrastructure.

Design of models and their parameters

Definition and construction a set of model configurations to call and run each model similarly. The (cgf-model-run) dictionary will store the model, its history, and the test datasets generated. The default model parameters are:

- n-steps: last 14 days

- n-horizon: next 7 days

- learning rate: 3e−4

The models also adopted a learning rate = 3e−4, which is a common learning rate and was found to be more desirable for each model during performance testing. Neural networks were built for the models to ensure efficient performance and accuracy of the results. To do this, improvements were added, and the parameters of each model were adjusted while defining (building its neural network). As shown:

DNN: A single 128-unit layer plus the common 128 and 24-unit layers with dropout.

CNN: Two Conv 1D layers with 64 filters each and kernel sizes of 6 and 3, respectively. After each Conv1D layer a maxpooling1D layer with size of 2.

LSTM: 2 LSTM layers with 72 and 48 units each.

CNN-LSTM: Combines the CNN and LSTM model, stack the CNN as input to the pair of LSTMs.

CNN-LSTM Skip Connection: Combines the CNN and LSTM layers with a skip connection to the common DNN layer.

CNN-LSTM Skip Attention: Combines the CNN and LSTM layers with a skip connection to the common DNN layer and adds an attention mechanism to the LSTM layers.

This shows that neural networks have multi-layered deep learning models, which are powerful because the multiple layers (recurrent neural networks) allow the models to learn from more complex patterns in the data used in the study.

Deep learning stepwise approach

Figure 4, appear the flow chart of the performance algorithm for the six deep learning models adopted in this study. This mechanism was built based on providing the best performance for each model, to obtain highly accurate predictive values by adding performance improvement functions and algorithms such as the ReLU (Rectified Linear Unit) activation function.

Figure 4 Chart of deep learning stepwise approach.

Models test

In Table 3, the summary of the models work shows the details of the preparation of the parameters used in the model’s performance (total, trainable and undefined).

Table 3 Models summary.

Model	Total params	Trainable params	Non-trainable params	
DNN	99,207	99,207	0	
CNN	49,671	49,671	0	
LSTM	54,023	54,023	0	
CNN-LSTM	85,735	85,735	0	
CNN-LSTM-Skip	117,991	117,991	0	
CNN-LSTM-Skip attention	160,999	160,999	0	

In Fig. 5, applying the predictions to the tested values and results of our tests, which will include forecasts for more than a year, will demonstrate the accuracy and consistency of our models. The predicted water levels in the six models are compared to observed water levels to measure their accuracy. The figure plots (actual) water levels (shown by the blue line) and (predicted) values (shown by the orange line) for about a year.

Figure 5 Normalized results.

In Fig. 6, the results in the model measure the performance of the DL model. It is determined by averaging the differences between all of the dataset’s data points’ actual and predicted values. The lower the value, the better the model works.

Figure 6 Performance metric results for DL models.

The graphical distribution of the figure represents actual water level data and our models’ forecasts for those levels. The figure shows a comparison between water levels as they are (blue line) and as predicted (orange line) over more than a year, and the horizontal axis (days cumulative) represents the vertical axis (water level behind the dam). Note that this is the required prediction.

Results

Accuracy of train models

Here, will count on the accuracy of the values resulting from that data and for each of the Eq. (6) trained models used in prediction in this study after those data were subjected to training, verification, and testing by calculating the loss and the mean absolute error (MAE).

Loss curve

In Fig. 7, the prediction loss curve is a graph showing us the relationship between the actual value and the expected value. It is used to evaluate the performance of a model. One must typically square the difference between the observed and predicted values to determine the loss curve. This resulting value is then averaged across all data points in the data set. A loss curve can be used to identify problems with a model, such as overfitting or underfitting. Since the ideal values in the loss curve used in the prediction are as close to zero as possible, notice in Fig. 3 above that the DL model works well and accurately predicts the values.

Figure 7 Loss curve.

MAE curve

The mean absolute error (MAE) curve is a graph that shows the relationship between MAE, and the number of training epochs. MAE measures the accuracy of a DL model. It is calculated by taking the absolute value of the difference between the predicted value and the actual value. MAE curve can identify problems with the DL model, such as overfitting or underfitting. The ideal values in the MAE curve are as close to zero as possible. So, notice in Fig. 8 that the DL model is performing well and accurately predicts the values.

Figure 8 MAE curve.

Comparison of the accuracy results of the train models

From the preceding above, review the outcome of the results for the accuracy of the values resulting from the work of each DL model, as shown in the Table 4

Table 4 Performance comparison of deep learning models.

Model	MAE	Error-MW	
DNN	0.097373	30.913406	
CNN	0.095262	30.243204	
LSTM	0.088246	28.015784	
CNN-LSTM	0.087153	27.668873	
CNN-LSTM-Skip	0.124239	39.442727	
CNN-LSTM-Skip attention	0.105618	33.530829	

Different performance levels may be seen in the deep learning models. MAE is a statistic used to judge how well a model’s predictions match the matching real-world data. MAE can be employed as a metric for assessing the efficacy of a model concerning specific tasks, such as machine translation or text summarization. A decrease in MAE suggests that the model is generating predictions with higher levels of accuracy.

Nevertheless, it is generally accepted that MAE below 10% indicates good performance, while an MAE below 5% is considered outstanding. Table 4 displays the lowest MAE value as 0.087153, associated with the hybrid cnn_lstm deep learning model. This indicates that the hybrid cnn_lstm deep learning model has the lowest average error when comparing projected and actual values. The graph illustrates the correlation between the observed and forecasted values for the hybrid cnn_lstm deep learning model. The x-axis represents the observed values, while the y-axis represents the expected values. The line depicted in the graph corresponds to the linear regression line, which is a line that optimally aligns with given data. The graph illustrates a high degree of proximity between the anticipated and actual values. Nevertheless, it is essential to acknowledge the presence of outliers, data points that deviate dramatically from the overall pattern. These outliers could be attributed to either model mistakes or data noise.

The CNN-LSTM model combines the performance of adversarial neural networks for processing visual data and the performance of recurrent neural networks for processing temporal data. It is a model characterized by multiple layers (iterations).

As a result, the CNN-LSTM model proved to be better than the other five models that accompanied it in this study, although the other five models also gave good results, good performance, and low and average error rates. This is due to the improvements made to the models’ performance, the clearing of noise, and the many outliers in the data.

Visualization of model comparisons

Figures 9 and 10 show the MAE and Error-MW values bar charts, as seen below. These charts visualize the model performance differences and supplement the information supplied in Table 4.

Figure 9 MAE models comparison.

Figure 10 Error-MW models comparison.

Bar charts in Figs. 9 and 10 give a different viewpoint on the model’s performance, enabling a rapid side-by-side comparison. A thorough examination of the model com-parison findings and the strengths and weaknesses of each model is made possible by combining the table and bar charts. Will explore the ramifications of these results in more detail and talk about potential directions for more study and development in the sections that follow.

It provided us with excellent results in the sense of mae, even though the other five models (DNN, CNN, LSTM, CNN-LSTM, CNN-LSTM-Skip, CNN-LSTM-Skip attention) demonstrated excellent results within the general perspective of the mae concept. Ensure precision and minimal mistake rates. This is because a memory (CNN-LSTM) model’s ability to predict statistical data depends on several variables, including the data’s complexity, the training dataset’s size and quality, and the model’s particular architecture and hyperparameters. (CNNs), which are good at collecting local information, are combined with LSTM networks that capture long-term dependencies to create hybrid deep learning models.

Evaluation and potential futures

The model comparison results offer insightful information on how well Amnesty International created the deep learning models for crisis management in the Mosul dam. Models that include a partnership between their different layers (hygienic) and multi-headed interest mechanisms, such as the hybrid CNN-LSTM deep learning model, show excellent prediction accuracy for the short-term target variable. The capacity of these models to capture short and long-term time allocations in time series data explains their efficacy. Front links allow the flow of first-hand information, allowing models to access pertinent historical data quickly. Additionally, the multi-heading method makes it easier to spot significant trends and patterns over a range of periods, improving models’ capacity for prediction. According to our measurement of the other models’ actual values, which shows how well they performed, the LSTM plus CNN combination significantly outperforms the other models in this situation. More research may be required to comprehend the details of the data and to investigate and enhance other model structures that more accurately depict the underlying patterns. The precision of CNN-LSTM emphasizes the significance of several architectural advancements. This model, together with its joint improvements to this model, bulk links, and multi-head interest, offers the possibility of achieving the most up to date performance in enterprise crisis management applications. Despite the encouraging findings, several caveats must be considered. The findings may have limited utility in other situations because of their specific dataset and associated tasks. Regarding crisis management at Mosul Dam, the evaluation employs artificial intelligence (AI) techniques. In addition, only deep learning methods have been considered for the model comparison; no other methods or ensemble techniques have been tested. Future studies may explore the potential inclusion of additional time series approaches that are not paramount to the present study. They may also investigate the influence of increased hyperparameters and how to combine methods to improve the predicted performance. Studying the interpretability of the models and the potential for adding knowledge domains can help us better understand the fundamental principles of time series dynamics.

In Fig. 11, an R2 value close to 1 is valid for models that seek to describe the relationship between variables. The values of R2 = 0.999 for the CNN model and R2 = 0.998 for the DNN model indicate a strong fit for these two models (Figs. 11A & 11B), and this value is the exact percentage of the relationship between the actual and predicted values within the scope of the current study. It is also noted that the R2 values for the other five models (DNN, CNN, LSTM, CNN-LSTM, CNN-LSTM-Skip, CNN-LSTM-Skip attention) are somewhat acceptable. It is essential to realize that the performance of the model depends on the specific scenario and data used. As a result, it is necessary to evaluate models using multiple metrics and consider the particular objectives.

Figure 11 (A–F) R2: regression of six models.

Conclusion

This study’s evaluation findings showed that CNN-LSTM outperformed five other models regarding precision, spotting meaningful temporal relationships, and precise target variable prediction. The hybrid CNN-LSTM deep learning model demonstrated strong performance and high predictability efficiency. The hybrid cnn_lstm deep learning model showed strong performance and high predictability efficiency for the future water level in the dam over 14 days before performing the calculations on it to predict seven days in advance, and that means that will take two weeks to predict the outcome within the next seven days, so will have time to manage our crisis and make the decision. It provided us with excellent results in the sense of MAE, even though the other five models (DNN, CNN, LSTM, CNN-LSTM, CNN-LSTM-Skip, CNN-LSTM-Skip attention) demonstrated excellent results within the general perspective of the MAE concept. Ensure precision and minimal mistake rates. This is because a memory (CNN-LSTM) model’s ability to predict statistical data depends on several variables, including the complexity of the data, the size and quality of the training dataset, and the model’s particular architecture and hyperparameters. CNNs, which are good at collecting local information, are combined with long short-term memory (LSTM) networks that capture long-term dependencies to create hybrid deep learning models.

The conclusion of this study is to predict the water level in the Mosul Dam, which suffers from problems in its foundations and may collapse in the future. It also aims to test the effectiveness of the six models proposed in this study after evaluating their performance and applying the prediction process within a scenario to obtain predictive values 14 days later. The results showed the practical effectiveness of the CNN-LSTM hybrid model in correctly and accurately obtaining prediction values within the desired scenario’s integrated framework with the lowest error rate.

Finally, the knowledge gathered from this study may be used to create reliable prediction models that can reduce risks and improve crisis readiness. Deploying emergency response personnel, equipment, and supplies to regions is at risk. Since this study is related to predicting the water level in the Mosul Dam in northern Iraq, using time series and deep learning models to manage the water level in the dam will significantly impact the challenges posed by potential climate change. By focusing on early warning systems, adaptive management, and improving the allocation of water resources, this study can contribute to ensuring dam safety, water security, and resilience in a changing climate.

For example, when rainfall increases or water releases by the State of Turkey on the Tigris River increases, the storage water level in the Mosul Dam will rise. Accordingly, these models will predict and alert monitoring centers of a rise in the water level, which may reach a dangerous height, and the possibility of a breach in the dam.

Thus, the current study is valuable in evaluating the extent to which deep learning models can help predict and mitigate such risks.

Supplemental Information

Supplemental Information 1 Code

Supplemental Information 2 Data

I would like to extend my thanks and appreciation to my country, Iraq, which supports me in my academic career and provided me with the necessary data to make the current article a success.

Additional Information and Declarations

Competing Interests

Author Contributions

Data Availability

The authors declare there are no competing interests.

Khalid MK Khafaji conceived and designed the experiments, performed the experiments, analyzed the data, performed the computation work, prepared figures and/or tables, authored or reviewed drafts of the article, and approved the final draft.

Bassem Ben Hamed conceived and designed the experiments, performed the experiments, performed the computation work, authored or reviewed drafts of the article, and approved the final draft.

The following information was supplied regarding data availability:

The code and data is available in the Supplemental Files.

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
