# Peer review of "Employing deep learning in crisis management and decision making through prediction using time series data in Mosul Dam Northern Iraq"

_PeerJ Computer Science, doi:10.7717/peerj-cs.2416_

## Round 0.1 · original submission · Major Revisions

The paper must be improved according to reviewers observations.

·

Basic reporting

This paper applied various deep artificial neural networks for predicting time series. The involved networks consisted of DNN, CNN, LSTM, and their combinations. The data are the 13 years’ worth of monthly water levels at the Mosul dam. It aims at utilizing the most suitable (CNN-LSTM) model to predict their level so that preventive measures could be prepared should a critical event occur. The paper is comprehensible despite many flaws. Although the implications and practical use case were not undoubtedly demonstrated, its scientific merits and technical content justify for publication. That being said, apart from the detailed structural comments given below, the paper could benefit from substantial language editing.
- Albeit extensive literature survey on time-series prediction of flood event were provided, that on dam water level analysis and relevant crisis and failure predictions was not. Please enhance. The author may kindly reduce redundant texts of the former, to make place for the latter.
- Section 3.2 are general overview of the field of study and seems out of place. It should be greatly reduced and moved to Section 1.
- The time series data of the monthly water level in the dam monitored over 13 years are provided in an easily accessible format. They enable the replication of the study.

Experimental design

- While a majority of the recent studies discussed in the introduction considered other geological, environmental and meteorological factors, the present study seems to analyse exclusively water levels. This approach is prone to overfitting (i.e., not in the statistical sense, but in the sense of generalizing the model to other areas). Please discuss and/ or provide supportive arguments.
- Basic descriptions on the CNN are LSTM architectures can be removed (citing relevant studies), while maintaining only those closely related to the methodology. In fact, specific configurations (hyper parameters, etc.) for each model are missing. Please consider.
- Since the analysed data was quite dated. Please explained why not newer data were considered and would the findings remain relevant in present time (with different environmental factors).
- Details on computing environments would be welcome.

Validity of the findings

- Please note that the not all the verbose results given by the AI software must be included in the manuscript. Kindly present only those relevant to key findings and conclusions.
- Descriptive and quantitative statistics of (water level) data are not yet provided. Please consider adding them in Section 3.1.
- Figure 7 is redundant and does not contribute greatly to the study value. Please kindly remove.
- The six DL networks were extensively assessed, and the relevant conclusions on CNN-LSTM’s superior (prediction) performance are valid and well supported. However, the same cannot be said about its prospective applications, i.e., warning against dam failure and thus proactive crisis management. More specifically, such scenario was neither proved nor supported by these data. There was, for instance, no historical event that the model would have been predicted well in advance. The author may provide such information in Section 2, e.g., pinpointing when the crisis did occur and prove in Section 4, whether it was anticipated by the model before, etc.

Additional comments

Please ensure that all numbers and text in the graphs are clearly visible both on printed pages and computer screen (of typical resolution). Currently many of them are too small and unsharp. Panel splitting, for example, may be employed to resolve this problem. Moreover, please also ensure that all the axes are titled (some of them are not), and their units are provided (most of them are not). Please also elaborate figure captions so that they are self-explanatory but brief. Finally, please ensure that variables, terms, notations, and abbreviations are generally accepted and used consistently within the text.

Reviewer 2 ·

Basic reporting

Interesting manuscript. However, the presented language has a major issue here. Please see the academic writing styles.

Experimental design

Why The ratio we will divide it by will be 80, 10, 10. That will be 3398.9 170 for practice, 850.9 for checking, and 500.9 for testing, as shown in Table 1?

Will be?
Again has some language issues!

Validity of the findings

Some of the figures are unclear. Please check all of them.
In addition, how did you look at the extreme values?

How is the compatibility of the models with respect to extreme values?

I would like to see square plots to R2 values. Please see the following papers.
https://www.mdpi.com/2076-3298/10/8/141

Additional comments

You will need to look at the latest research work on XAIs. At least to state the importance of XAIs in the application.

In addition, what is the impact of potential climate change?

Reviewer 3 ·

Basic reporting

1. The paper contains several grammatical errors and unclear sentences that hinder comprehension. It is recommended to proofread the paper thoroughly to improve language quality and clarity.

2. The literature review is insufficient and does not provide a comprehensive background. Expand the literature review to include recent studies and theories, and discuss how the current research relates to existing knowledge.
Chen et al. (2023) https://doi.org/10.5194/gmd-16-2915-2023 discuss an advanced hydrological-geotechnical model for prediction, enriching the discussion on prediction accuracy and model efficiency. Zhang et al. (2023) https://doi.org/10.1029/2023GL104324 and Zhao et al. (2024) https://doi.org/10.1175/JCLI-D-23-0400.1 explore precipitation processes and mechanisms, adding depth to the understanding of regional climatic factors relevant to the Mosul Dam scenario. Du and Wang (2013) doi: 10.1785/0120120185 focus on spatial correlations for seismic activities, enhancing the geotechnical aspects of the model. Zhu (2023) doi: 10.33168/JLISS.2023.0309 introduces an adaptive decision model using deep reinforcement learning, potentially improving predictive accuracy. Ye et al. (2024) https://doi.org/10.1016/j.jhydrol.2024.130905 provide insights into long-term deformation monitoring, aligning with the paper's focus on the Mosul Dam. Liu et al. (2024) https://doi.org/10.1016/j.measurement.2024.114999 research fault diagnosis using time series segmentation, which can inspire similar techniques for hydrological models. Hu et al. (2024) https://doi.org/10.1061/JPSEA2.PSENG-1453 predict surface settlement using machine learning, introducing additional methods to enhance predictive capabilities. Yang et al. (2022) doi: 10.3390/rs14020303 discuss graph convolutional network, applicable to spatial-temporal analysis in the paper. Chen et al. (2024) https://doi.org/10.1016/j.oregeorev.2024.106030 use fractal theory and machine learning for metallogenic prediction, offering novel perspectives for better prediction outcomes. Yin et al. (2023) https://doi.org/10.1016/j.ecolind.2023.110837 analyze precipitation and river discharge during the operation of the Three Gorges Dam, providing a basis for comparative analysis. Qin et al. (2024) https://doi.org/10.1007/s12555-022-0104-x introduce an RCLSTMNet for forecasting, potentially enhancing the hybrid CNN-LSTM models discussed in the paper. By integrating these references, the paper will benefit from a richer literature foundation, reinforcing the validity and relevance of its findings within the broader context of current research.

3. The structure of the article is mostly professional, but some sections lack clarity. Improve the quality and clarity of visualizations, ensuring they are well-labeled and explained.

4. The results section is generally self-contained, but the analysis of the results needs to be more thorough. Offer a more detailed analysis, discussing the strengths and weaknesses of each model.

Experimental design

1. The research question is well defined and relevant, but the paper should better articulate how it fills the identified knowledge gap. Add a clear statement of objectives and scope in the introduction.

2. The investigation appears rigorous, but there is no discussion on ethical considerations. Include a section discussing ethical implications, including data privacy and potential biases.

3. The methodology section lacks sufficient detail. Provide a more detailed description of the data preprocessing, model parameters, and validation techniques.

Validity of the findings

1. The conclusions are well stated but could be strengthened by linking them more explicitly to the original research question and supporting results.

2. Provide more detailed results analysis and baseline model comparisons to highlight improvements.

Additional comments

The paper presents promising findings, but several areas need improvement. Discuss the interpretability of the models, address ethical considerations, and provide more detailed visualizations. Additionally, consider practical applications and future research directions to enhance the paper's contribution to the field.

Reviewer 4 ·

Basic reporting

The paper language over all is written in acceptable English proficiency, but does not follow the scientific writing approach with avoiding the use of ‘we’ in the text and focusing on the research not the person. Also, there is poor use of proper present and past verb tenses, that does not reflect the current status of the research stage.
As for the structure of the research it is sound and the LR is relevant and sufficient, well presented as well, all the figures and data are made available, although the figure numbers are missed in the text few times.
The research tool is correct, the solution is applicable, and however the data are not sufficient to get the required results, since the data is not continuous and therefore can’t be used to train neural network to forecast results with more than 20 years data gap.

Experimental design

The research is relevant and important, the knowledge gap is accurate, and the problem addressed is current and urgent.
Moreover, the research tool is applicable and appropriate to address the research approach. The research method is well explained.

Validity of the findings

The research is very impactful and the tool can solve a real and urgent problem.
The contribution and significance are clearly stated and well explained. Moreover, the research approach and the proposal of using neural network are excellent choice to solve a forecasting problem. Also, the error analysis measures used are sound and sufficient.
However, the data used are not continues and have a major gap that makes this time series unfit to train the network for current forecast, the data patterns as well as the time series behavior have changed through the data gap and the disconnected data can’t be used to train the network, therefore the results and findings are not usable, moreover the results can’t be used to validate the results or the approach. Consequently making the findings invalid and not usable.
Moreover, the damage to the Mosul Dam has happened through the years and with no data to know the actual status of the Dam and the water capacity allowable and how it changed through the year, there is no way to validate the change in the behavior of the time series. And the data used in the training are not similar to the current data. Therefore, although the proposed method is good and applicable but the results and findings are not valid and not significant. Therefore, cant be used for making critical decision on managing the Dam crisis.

Additional comments

It would be very useful for the reader understanding and the research visibility if the research method is to be presented in terms of a figure , such as a flowchart or process diagram.

·

Basic reporting

The findings in Figure no. 3 are not clear; please specify in detail the description.

Draw Figures 3, 4, 5, and 6 again to increase their visibility.

The study area should be explored to clearly specify the objective of the research. There is no novelty in the paper.
Only 11 references are studied, which are not the latest and most up-to-date. Current work in this area should be studied and referred to

In the introduction part, the existing work are described and referenced without a proper logic. A more targeted overview and summary according to the topic of this paper need to be conducted again.
Please discuss the limits of the literature and explain how you fill some of the existing gaps. Limited resources have been used and need to be developed and supplemented.

The most important basic researches for this research along with mentioning the identified gaps and their innovations and the criteria considered should be compared with the present research in a table

Experimental design

It is not clear how research work uses the LSTM model in the study to improve accuracy.
Please specify the references of equations and mathematical expressions used in the study.

There is no contribution in terms of the proposed model. There is a lack of mention of the research gap.

Validity of the findings

The robustness of the model should be checked.

The objective and impact of doing this work is not highlighted.
The presentation of results is not satisfactory. Figures in the Results section are not readable, hindering a clear understanding of the outcomes.

---

## Round 0.2 · Minor Revisions

The authors need to revise the paper since there are minor revisions.

·

Basic reporting

The authors have satisfactorily addressed my concerns. The academic contribution of the paper has merit for publishing in PeerJ, subject to the following conditions being fulfilled.
1. Authors responses (only rebuttons) to my comments are appropriately rephrased and incorporated into relevant places of the revised manuscript, so that the readers are made aware of the arguments as well as the current limitations.
2. Editorial quality including the graphs (resolution, clarity, appearance, etc.) and texts (academic writing style and language) are properly enhanced (probably during the production stage).

Experimental design

no further comment

Validity of the findings

no further comment

Additional comments

no further comment

Reviewer 2 ·

Basic reporting

Where are your R2 plots?
See the reference which I suggested. It has clearly shown the R2 plots in square plots in scales.

Experimental design

NA

Validity of the findings

NA

Additional comments

NA

Reviewer 3 ·

Basic reporting

The authors have thoroughly addressed all the suggested corrections and revisions, including grammatical improvements, an expanded literature review, and enhanced visualizations and analysis. The paper is ready for acceptance.

Experimental design

The authors clarified the study's objectives in the introduction and added detailed descriptions of data preprocessing and model parameters in the methodology section.

Validity of the findings

The authors provided a more detailed analysis of model performance, clearly demonstrating the strengths of each model.

Additional comments

The authors have successfully addressed all comments, improving clarity, methodology, and analysis. The paper is now suitable for acceptance.

---

## Round 0.3 · accepted · Accept

The paper can be accepted. It was well improved.